# Metabolic Profiling by GC-MS, *In Vitro* Biological Potential, and *In Silico* Molecular Docking Studies of *Verbena officinalis*

**DOI:** 10.3390/molecules27196685

**Published:** 2022-10-08

**Authors:** Rabia Nisar, Saeed Ahmad, Kashif-ur-Rehman Khan, Asmaa E. Sherif, Fawaz Alasmari, Afaf F. Almuqati, Chitchamai Ovatlarnporn, Mohsin Abbas Khan, Muhammad Umair, Huma Rao, Bilal Ahmad Ghalloo, Umair Khurshid, Rizwana Dilshad, Khaled S. Nassar, Sameh A. Korma

**Affiliations:** 1Department of Pharmaceutical Chemistry, Faculty of Pharmacy, The Islamia University of Bahawalpur, Bahawalpur 63100, Pakistan; 2Department of Chemistry and Physics, College of Sciences and Mathematics, Arkansas State University, Jonesboro, AR 72404, USA; 3Department of Pharmacognosy, College of Pharmacy, Prince Sattam bin Abdul Aziz University, Alkharj 11942, Saudi Arabia; 4Department of Pharmacognosy, Faculty of Pharmacy, Mansoura University, Mansoura 35516, Egypt; 5Department of Pharmacology and Toxicology, College of Pharmacy, King Saud University, Riyadh 11451, Saudi Arabia; 6Department of Pharmaceutical Chemistry, College of Pharmacy, University of Hafr Al-Batin, Hafr Al-Batin 31991, Saudi Arabia; 7Department of Pharmaceutical Chemistry, Faculty of Pharmaceutical Sciences, Prince of Songkla University, Hat Yai 90110, Thailand; 8Department of Food Science and Engineering, College of Chemistry and Engineering, Shenzhen University, Shenzhen 518060, China; 9Department of Food, Dairy Science and Technology, Faculty of Agriculture, Damanhour University, Damanhour 22516, Egypt; 10Department of Food Science, Faculty of Agriculture, Zagazig University, Zagazig 44519, Egypt; 11School of Food Science and Engineering, South China University of Technology, Guangzhou 510641, China

**Keywords:** *Verbena officinalis*, natural compounds, flavonoids, polyphenols, GC-MS, antioxidant, chronic diseases, hemolytic activity, molecular docking

## Abstract

*Verbena officinalis* L. is a traditionally important medicinal herb that has a rich source of bioactive phytoconstituents with biological benefits. The objective of this study was to assess the metabolic profile and in vitro biological potential of *V. officinalis*. The bioactive phytoconstituents were evaluated by preliminary phytochemical studies, estimation of polyphenolic contents, and gas chromatography-mass spectrometry (GC-MS) analysis of all fractions (crude methanolic, *n*-hexane, ethyl acetate, and *n*-butanol) of *V. officinalis*. The biological investigation was performed by different assays including antioxidant assays (DPPH, ABTS, CUPRAC, and FRAP), enzyme inhibition assays (urease and α-glucosidase), and hemolytic activity. The ethyl acetate extract had the maximum concentration of total phenolic and total flavonoid contents (394.30 ± 1.09 mg GAE·g^−1^ DE and 137.35 ± 0.94 mg QE·g^−1^ DE, respectively). Significant antioxidant potential was observed in all fractions by all four antioxidant methods. Maximum urease inhibitory activity in terms of IC_50_ value was shown by ethyl acetate fraction (10 ± 1.60 µg mL^−1^) in comparison to standard hydroxy urea (9.8 ± 1.20 µg·mL^−1^). The *n*-hexane extract showed good α-glucosidase inhibitory efficacy (420 ± 20 µg·mL^−1^) as compared to other extract/fractions. Minimum hemolytic activity was found in crude methanolic fraction (6.5 ± 0.94%) in comparison to positive standard Triton X-100 (93.5 ± 0.48%). The GC-MS analysis of all extract/fractions of *V. officinalis* including crude methanolic, *n*-hexane, ethyl acetate, and *n*-butanol fractions, resulted in the identification of 24, 56, 25, and 9 bioactive compounds, respectively, with 80% quality index. Furthermore, the bioactive compounds identified by GC-MS were analyzed using in silico molecular docking studies to determine the binding affinity between ligands and enzymes (urease and α-glucosidase). In conclusion, *V. officinalis* possesses multiple therapeutical potentials, and further research is needed to explore its use in the treatment of chronic diseases.

## 1. Introduction

The existence of a wide range of secondary metabolites in medicinal plants has led to extensive investigation of these plants in recent years to identify the lead compound that can contribute to the management of chronic diseases and therapeutic effectiveness [1]. There has been an increase in scientific interest in medicinal plants [2]. Natural products have been used therapeutically to cure many diseases since ancient times. According to the World Health Organization, 80% of people around the world use plant-based treatments to cover their basic health needs [3]. Approximately 52% of approved molecules from 1981–2014 were natural products or derived directly from them [4]. According to multiple studies, secondary metabolites that are separated from medicinal plants are responsible for a variety of therapeutic uses, including antioxidant, antibacterial, anti-inflammatory, antiviral, antifungal, and anticancer [5]. Phytomedicines have been used largely due to their safety, accessibility, low cost, and sociological acceptance when compared to synthetic drugs [6,7].

Research on enzyme inhibition has expanded significantly over the last two decades [8]. Urease is a metalloenzyme containing nickel that facilitates the rapid conversion of urea into ammonia along with carbon dioxide. Urease is abundantly found in numerous plants, a variety of bacteria, and selected fungi [9,10]. One of the virulent elements in the pathogenesis of the gram-negative, microaerophilic, stomach-found *Helicobacter pylori* is ureases. The *H. pylori* infection can cause gastrointestinal inflammation, which raises the risk of chronic disorders, such as duodenal and gastric ulcers, gastric adenocarcinoma, and gastric lymphoma [11,12,13]. Researchers are being encouraged to find new urease inhibitory compounds from natural resources because the blocking of ureases is thought to be the most successful treatment for urease-dependent bacterial infections [14]. So, the discovery of safe and effective urease inhibitors is a demand nowadays due to the release of urease by microorganisms in different pathological disorders [15].

The enzyme α-Glucosidase, found on the intestinal cell membrane surface, catalyzes the breakage of α-glycosidic linkage present in oligosaccharides to make monosaccharides. Hence, inhibitors of α-glucosidase can postpone the generation of d-glucose from complex carbohydrates thus slowing down glucose absorption, and lowering the level of postprandial plasma glucose [16]. To decrease disorders associated with diabetes, regulating the concentration of glucose is a primary technique [17]. The incidence of postprandial hyperglycemia among diabetic individuals is reduced by the inhibitory activity of α-glucosidase, which is thought to interfere with the digestive process of carbohydrates. Several inhibitors of α-glucosidase, such as miglitol and acarbose, have been discovered [18]. However, acarbose use has been associated with gastrointestinal disturbances [19]. The ability of natural products to block the activity of digestive enzymes, and hence lower hyperglycemia in the management of chronic diabetes, had been successfully demonstrated by numerous researchers [20].

*Verbena officinalis* L. (Verbenaceae) is known as the herb of grace, pigeons’ grass and vervain. It is primarily found in North Africa, Asia and all over Europe. It is mostly distributed in wastelands and near water in cultivated fields in the northern as well as western regions of Pakistan. *V. officinalis* is a perennial erect small herb that grows up to 25–100 cm in height with lobed and serrated leaves. Pink or purple is the color of flowers [21]. *V. officinalis* has been utilized to alleviate several ailments in the folk medicinal system, including rheumatic pain, thyroid problem and wounds [22], gastric diseases, skin burns, abrasion [23], cough and asthma [24], depression, amenorrhea, and acute dysentery enteritis [25]. *V. officinalis* has been studied for its important bioactivities such as antioxidant [26], diuretics and expectorant analgesic and anti-inflammatory [27,28], anticonvulsant [21], antifungal [29], antibacterial [30,31], anticancer [32,33], antidepressant [34], neuroprotective [25], urolithiasis [35] antiproliferative [36] and antitumor [37] effects. The abundance of bioactive metabolites in *V. officinalis*, including flavonoids [38], phenylethanoid glycosides [36], sterols and triterpenoids [39], and ursolic acid [40] explains the folklore use of *V. officinalis* [21].

In account of this, a crude methanolic extract of *V. officinalis* (CRVO) was prepared, then fractionated using various solvents in ascending order of their polarity to produce different fractions; *n*-hexane (NHVO), ethyl acetate (EAVO) and *n*-butanol (NBVO). The methanolic crude extract along with its various fractions was evaluated for its total phenolic content (TPC) and total flavonoid content (TFC), urease and α-glucosidase inhibition assays, and antioxidant assays by different methods (DPPH, ABTS, CUPRAC, and FRAP). Metabolic profiles of all fractions of the whole plant of *V. officinalis* were performed by gas chromatography-mass spectrometry (GC-MS) to identify the tentative secondary metabolites in the respective fractions. In silico molecular docking studies were also conducted for the bioactive compounds identified in all fractions by GC-MS.

## 2. Results

### 2.1. Phytochemical Profile of V. officinalis

#### 2.1.1. Preliminary Phytochemical Assessments

Preliminary phytochemical testing of CRVO, NHVO, EAVO, and NBVO fractions of the whole plant of *V. officinalis* were performed. This analysis confirmed the presence of many bioactive primary and secondary metabolites, as shown in Table 1. The plant showed the presence of primary and secondary plant bioactive metabolites, including carbohydrates, saponins, tannins, phenols, flavonoids, starch, alkaloids, glycosides and resins.

#### 2.1.2. Polyphenolic Contents Estimation

Total phenolic content (TPC)

The maximum amount of TPC was observed in EAVO (394.30 ± 2.50 mg GAE·g^−1^ DE) and the minimum amount was observed in CRVO (89.07 ± 1.88 mg GAE·g^−1^ DE) (milligram gallic acid equivalent per gram weight of dry extract) (Figure 1).

Total flavonoid content (TFC)

The EAVO sample showed the highest amount of TFC with a value of 137.35 ± 0.94 mg QE·g^−1^ DE (milligram quercetin equivalent per gram weight of dry extract) and the CRVO sample exhibited the lowest amount of TFC with a value of 66.26 ± 1.42 mg QE·g^−1^ DE (Figure 2).

#### 2.1.3. Detection of Bioactive Compounds by GC-MS

CRVO, NHVO, EAVO, and NBVO fractions of the whole plant of *V. officinalis* were subjected to GC-MS analysis. The mass spectra of each plant metabolite at different retention times were checked with databases of the mass spectra of the National Institute Standard and Technology (NIST-14). Tentatively identified compounds in CRVO, NHVO, EAVO and NBVO fractions of *V. officinalis* were 112, 112, 90, and 46, respectively. Compounds with a quality index of more than 80% were finally selected and represented in Appendix A. The retention time in minutes (RT), peak area (%, calculated by dividing each compound peak area by the sum of all compounds’ peak areas within the sample), name of the compound, molecular formula, and molecular weight of the metabolites identified in CRVO, NHVO, EAVO, and NBVO fractions of *V. officinalis* using GC-MS were shown in Appendix A. The GC-MS chromatogram of CRVO, NHVO, EAVO, and NBVO fractions of *V. officinalis* were exhibited in Appendix A.

### 2.2. In Vitro Biological Investigation of V. officinalis

CRVO, NHVO, EAVO, and NBVO fractions of the whole plant of *V. officinalis* were evaluated for their biological potential using different approaches such as antioxidant assay, enzyme inhibition and hemolytic activities.

#### 2.2.1. Antioxidant Assays

Radical scavenging potential

The radical scavenging potential of the whole plant of *V. officinalis* was determined using DPPH and ABTS methods. The order of activity of different fractions was as follows; EAVO > NBVO > CRVO > NHVO for DPPH and EAVO > NBVO > CRVO > NHVO for ABTS. The highest scavenging potential estimated by the DPPH method was shown by EAVO (161.21 ± 2.02 mg TE·g^−1^ DE), and the minimum value was shown by NHVO (34.30 ± 2.02 mg TE·g^−1^ DE). The maximum free radical scavenging activity calculated by the ABTS method was shown by EAVO (178.57 ± 0.83 mg TE·g^−1^ DE), while the lowest value was observed in NHVO (51.77 ± 0.36 mg TE·g^−1^ DE) (Figure 3).

Reducing power antioxidant assay

The reducing antioxidant potential of CRVO, NHVO, EAVO and NBVO fractions of the whole plant of *V. officinalis* was evaluated by two methods namely CUPRAC and FRAP assays. The results were as follows: EAVO > NHVO > NBVO > CRVO for CUPRAC assay and EAVO > NBVO > NHVO > CRVO for FRAP assay. EAVO showed the maximum reducing potential with a value of 592.88 ± 2.44 mg TE·g^−1^ DE and CRVO showed the minimum reducing capacity value of 223.11 ± 1.55 mg TE·g^−1^ DE for CUPRAC assay. The highest reducing activity was calculated for FRAP assay for EAVO with a value of 360.18 ± 2.68 mg TE·g^−1^ DE, and the lowest was determined for CRVO, with a value of 152.171 ± 2.68 mg TE·g^−1^ DE (Figure 3).

#### 2.2.2. *In vitro* Enzyme Inhibition Assay

Urease inhibition assay

The urease inhibitory potential of CRVO, NHVO, EAVO, and NBVO fractions of *V. officinalis* whole plant was evaluated using a previously modified method [41]. Results were presented as IC_50_ values. Lower IC_50_ values indicate the highest enzyme inhibition. The order of inhibition of the urease enzyme of crude methanolic extract and different fractions of the whole plant of *V. officinalis* were as follows; EAVO < NBVO < NHVO < CRVO. The IC_50_ values of EAVO and NBVO were found to be 10 ± 1.60 µg·mL^−1^ and 30 ± 2.40 µg·mL^−1^ in comparison to IC_50_ value of hydroxy urea 9.8 ± 1.20 µg·mL^−1^. The results of urease inhibitory activity of different fractions of the whole plant of *V. officinalis* showed the plant as a potent inhibitor of the urease enzyme (Table 2).

α-Glucosidase inhibition assay

α-Glucosidase enzyme inhibition assay results were expressed as IC_50_ values. NHVO showed the best IC_50_ value of 420 ± 20 µg·mL^−1^ with good antidiabetic potential while EAVO showed a moderate IC_50_ value of 685 ± 31 µg mL^−1^ and IC_50_ for quercetin was 10 ± 1.30 µg·mL^−1^ (Table 2).

#### 2.2.3. Hemolytic Activity

The hemolytic potential of CRVO, NHVO, EAVO, and NBVO fractions of the whole plant of *V. officinalis* were exhibited (Table 3). The percentage of hemolytic activity was found in the order: NBVO > EAVO > NHVO > CRVO. The maximum hemolytic value was 14.5 ± 1.20% for NBVO and the minimum hemolytic value was 6.5 ± 0.94% for CRVO. Results confirmed that *V. officinalis* is a safe and non-toxic plant due to less than 30% of hemolysis activity [1].

### 2.3. In Silico Molecular Docking Studies

Molecular docking studies were performed for both urease (PDB DOI: 10.2210/pdb1E9Z/pdb) and α-glucosidase (PDB DOI: 10.2210/pdb5ZCB/pdb). All the compounds from GC-MS profiles of methanolic, *n*-hexane, ethyl acetate, and *n*-butanol fractions were docked against urease and α-glucosidase enzymes. Four compounds showed the best binding affinity against both enzymes. Benzenepropanoic acid, 3,5-bis(1,1-dimethylethyl)-4-hydroxy-, methyl ester showed the maximum binding affinity i.e., −6.8 Kcal·mol^−1^ against urease and α-glucosidase. ar-Turmerone showed a −5.8 Kcal·mol^−1^ binding affinity against urease, and it showed a −6.5 Kcal·mol^−1^ binding affinity against α-glucosidase. Curlone showed a −5.6 Kcal·mol^−1^ binding affinity against urease while it showed a −5.9 Kcal·mol^−1^ binding affinity against α-glucosidase. 3-pyrazolidinone, 4,4-dimethyl-1-phenyl has a binding affinity of −5.7 Kcal·mol^−1^ against urease and −5.8 Kcal·mol^−1^ against α-glucosidase while the binding affinity of hydroxy urea and quercetin (standards) for these enzymes (urease and α-glucosidase) was −4.1 and −7.9 Kcal·mol^−1^, respectively.

The molecular docking study was validated by redocking of urease, α-glucosidase, and selected ligands with Autodock-1.5.6. Additionally, the same results were found in terms of the binding affinity and RMSD values. The docking results of the four ligands with both enzymes are depicted in Table 4 and Figure 4 and Figure 5.

## 3. Discussion

Plants naturally contain substances called phytochemicals, which can have beneficial or harmful effects on human health [42]. The most abundant biological reservoirs of different phytochemicals are medicinal plants, which are used to treat various diseases and conditions. The biological potential of plants may be due to the presence of metabolites [43]. Alkaloids, phenolics, flavonoids, saponins, tannins, steroids, terpenoids, terpenes, glycosides, coumarins, polysaccharides, and other significant secondary metabolites can be found in plants [43]. Major phytochemicals including alkaloids having antimicrobial and analgesic potential, flavonoids and tannins demonstrated antioxidant and antibacterial activity [44], and saponins have anticancer, anti-inflammatory, antibacterial, and anti-diabetic potential [45]. The phytochemical analysis of *V. officinalis* extracts exhibited plant has a variety of phytochemicals such as phenols, flavonoids, glycosides, alkaloids tannins, saponins, resins, and terpenes. Previous studies showed the presence of these chemicals in the leaves of *V. officinalis* [46]. The results suggested that the medicinal activity of *V. officinalis* may be attributable to the presence of these phytoconstituents.

Polyphenols are bioactive substances that are frequently found in food products made from plants, such as fruits, seeds, and cereals. Polyphenols help to prevent the possibility of chronic diseases including cancer, neurodegenerative and cardiovascular diseases [47]. Several important flavonoids were reported in *V. officinalis* previously [48]. Flavonoids were reported to possess different biological activities, including antioxidant, anticancer, anti-inflammatory, antiviral, hepatoprotective, anti-fungal, and antibacterial [49]. The ethyl acetate fraction showed the highest values of both phenolic and flavonoid contents with values of 394.30 ± 1.09 mg GAE·g^−1^ DE and 137.35 ± 0.94 mg QE·g^−1^ DE, respectively while the crude methanolic fraction showed the minimum values of both phenolic and flavonoid contents having values of 89.07 ± 1.88 mg GAE·g^−1^ DE and 66.26 ± 1.42 mg QE·g^−1^ DE, respectively. Previous studies reported the presence of flavonoids (in hydroalcoholic and aqueous extracts, 0.76 and 0.79 g/100 g, respectively) and phenolic constituents (in hydroalcoholic and aqueous extracts, 1.25 and 1.75 g/100 g, respectively) in *V. officinalis* aerial parts contributing their role in the antioxidant potential of *V. officinalis* [50].

Oxidation processes are essential to provide the energy to support biological activity in living organisms. As a result, the uncontrolled production of oxygen reactive species (ROS) is associated with many chronic diseases, including cancer, atherosclerosis, rheumatoid disease, and degenerative processes linked with aging [51]. Synthetic as well as semi-synthetic antioxidants are frequently employed to reduce ROS damage, however, they have also been linked to cancers and damage to cells or entire organs (such as the liver) [51]. As a result, there is a substantial need for natural and functional antioxidants that can lower ROS overproduction and stop the progression of many chronic diseases. Natural antioxidants are considered safer as compared to synthetic antioxidants [52]. By neutralizing ROS, natural antioxidants obtained from plants are particularly effective at preventing the oxidation process. Bioactive compounds from plants exert their antioxidant activity via multiple mechanisms, including activation of Nrf_2_/ARE (Nuclear factor erythroid 2-related factor 2/Antioxidant response element) and deactivation of the NF-кB (Nuclear factor kappa B) pathways, directly involved in the inflammatory reaction [53]. Additionally, drugs made from plant sources are thought to be safer than synthetic ones [54].

The antioxidant activity of *V. officinalis* plant extracts has been established in various scientific studies, which is essential in the prevention of heart disease and cancer [55]. A study from the Faculty of Pharmacy of the University of Navarra in Spain on the antioxidant potential of 50% ethanolic as well as an aqueous extract of the plant proved beneficial in the removal of free radicals. The DPPH assays revealed that both extracts had substantial antiradical activity. The IC_50_ was 21.04 ± 1.61 µg·mL^−1^ and 33.8 ± 0.43 µg·mL^−1^ for ethanolic and aqueous extract, respectively. Xanthan oxidase is an enzyme that induces the production of oxygen radicals and was likewise inhibited by the solutions. The fraction including verbascoside and small quantities of luteolin 7-glucoside, isoverbascoside, and 1,5- and 4,5-dicaffeoylquinic acid had the highest antioxidant activity [50]. Another study performed in the College of Pharmacy; Woosuk University (Korea) revealed that methylene chloride fraction showed strong scavenging potential on DPPH radical, nitric oxide radical, superoxide radical, and 2,2’-azino-bis(3-ethylbenzthiazoline-6-sulphonic acid) radical exhibiting its potent reducing effect [55]. Research conducted in Spain exhibited the antioxidant effect of different fractions obtained from 50% methanolic extract of *V. officinalis* as well as some compounds isolated from this plant [29]. The amount of polyphenols have been found to directly correlate with scavenging capability [56], and EAVO showed a greater TPC and TFC (394.30 ± 2.50 mg GAE·g^−1^ DE and 137.35 ± 0.94 mg QE·g^−1^ DE), respectively. As a result, EAVO may be a source of free radical scavengers that naturally combat high ROS burdens. Phenolic compounds demonstrate redox characteristics, with the ability to act as antioxidants [57]. There was no comprehensive study reported on the antioxidant activity of the different solvents i.e., methanolic extract, *n*-hexane fraction, ethyl acetate fraction and *n*-butanol fraction of the whole plant of *V. officinalis*.

*Helicobacter pylori* is one of the causes of dyspepsia and extra-digestive problems linked with peptic ulcers worldwide [58,59]. The World Health Organization also validated this, stating that *H. pylori* is a class one carcinogen for gastric carcinoma and that it was determined that carcinogenic infections, including *H. pylori*, were responsible for 12% of malignancies detected in 2012. The success of synthetic drugs to cure gastric ulcers is overshadowed due to the toxicity risks associated with such drugs. Additionally, *H. pylori* resistance to antibiotics was among the list of antibiotic-resistant major priority diseases that encouraged researchers to find new antibiotics to eradicate this pathogen [60,61]. Given this, the goal of this study was to evaluate the urease inhibitory potential of *V. officinalis* as there is no comprehensive literature found on the urease inhibitory activity of the whole plant of *V. officinalis*. EAVO and NBVO fractions exhibited the highest urease inhibitory potential with IC_50_ of 10 ± 1.60 µg·mL^−1^ and 30 ± 2.40 µg·mL^−1^, respectively as compared to hydroxy urea IC_50_ value of 9.8 ± 1.20 µg·mL^−1^. The NHVO and CRVO fractions showed moderate results with IC_50_ values of 324 ± 16.40 µg·mL^−1^ and 465 ± 20.20 µg·mL^−1^, respectively (Table 2). The IC_50_ value for Urease inhibition of *Terminalia neotaliala* different extract/fractions is 1.79−3.54 mg·mL^−1^ [62]. The different extract/fractions of *Rondeletia odorata* at a concentration of 5 mg·mL^−1^ revealed urease inhibition of 45.69–73.39% [1]. The significant urease inhibitory activity of EAVO may be validated due to some bioactive constituents found in GC-MS of ethyl acetate fraction such as 9,12-Octadecadienoic acid (Z,Z)-, methyl ester [63], 2-Cyclopenten-1-one, 3-methyl- [64]. Several naturally available flavonoids including (quercetin), flavones, isoflavone, and polyphenolic compounds showed promising urease inhibitory activity [65,66,67]. Polyphenols and flavonoids exhibited antioxidant activity associated with anti-ulcer activity due to the production of free radicals in gastric mucosal abrasions. Histological data confirmed that the highest flavonoid contents in fraction might be involved in significant inhibition of the generation of reactive radical species indicating their role in gastric protection with anti-oxidant potential [68,69]. The current study revealed that *V. officinalis* is rich in flavonoids and phenols. Until now, there has been no comprehensive research on the urease inhibitory activity of the whole plant of *V. officinalis*. Further research on this plant might result in its use as a potent inhibitor of urease.

Diabetes mellitus is accompanied by hyperglycemia which has other consequences, including retinopathy, neuropathy, nephropathy, atherosclerosis, and cardiac dysfunction etc. Additionally, the glycation of several proteins may be brought on by hyperglycemia and result in chronic dysfunctions. Around 28,000 plant species have been documented for their therapeutic properties throughout the world, and approximately 3000 plant species, have the ethnopharmacological potential to manage diabetes and other problems [70]. The NHVO showed the promising inhibition of α-glucosidase with an IC_50_ value of 420 ± 20 µg·mL^−1^ when compared to quercetin with an IC_50_ value of 10 ± 1.30 µg·mL^−1^. EAVO showed moderate results of α-glucosidase inhibition with an IC_50_ value of 685 ± 31 µg mL^−1^ (Table 2). The IC_50_ value for α-glucosidase inhibition of *Terminalia neotaliala* different extract/fractions is 210−730 µg·mL^−1^ [62]. The α-glucosidase inhibition potential of NHVO was verified by GC-MS analysis by the presence of thymol [71], Neophytadiene [72], and ar-Turmerone [73]. This was the first time to study the different fractions of the whole plant of *V. officinalis* for antidiabetic potential. So, it’s important to perform further testing to determine which compounds are safe and efficient for managing diabetes.

Hemolysis, which results in the release of hemoglobin from red blood cells (RBCs), is the dissolution or breakage of the integrity of the RBC membrane [74]. The prolonged usage of some traditional plants can cause a potential toxic effect [75]. Many plants possess chemical constituents that could either hemolyze or anti-hemolyze activity on human RBCs. Plant extracts have the potential to disrupt red blood cell membranes resulting in harmful adverse effects, including the development of hemolytic anemia. Therefore, it is necessary to assess the potential hemolytic activity of several of the regularly utilized plants [76]. The plant extracts are considered dangerous to erythrocytes if there is more than 30% hemolysis [1].

The hemolytic activity of different fractions of the whole plant of *V. officinalis* was presented in Table 3. Results showed that CRVO possesses the minimum hemolytic percentage (6.5 ± 0.94%), whereas NBVO possesses maximum hemolytic activity (14.5 ± 1.20%). All fractions have hemolysis activity of less than 30% so all the fractions are safe and non-toxic to humans. This is the first time to report hemolytic activity of the whole plant of *V. officinalis*.

Additionally, it is possible to make significant advancements to in vitro research techniques for the quick screening of enzyme inhibitors utilizing molecular modeling. Therefore, to assess the biological activities of the extract and fraction, a combination of bioinformatics simulation and in vitro study will be helpful. Docking is a method of molecular modeling used to foretell how proteins (enzymes) will interact with small molecules (binders or ligands) [77]. Therefore, a thorough comprehension of protein-ligand interactions is essential to comprehending biology at the molecular level. Additionally, understanding the mechanisms underlying the interactions and binding between proteins and ligands can help in the discovery, design, and creation of pharmaceuticals. The binding affinity plays a crucial role in the interaction between ligands and enzymes. The better the interaction between the ligands and enzyme, the lower the binding affinity. The absence of contact between the ligand and the enzyme is represented by the binding affinity’s positive (+) sign. To gain a better understanding of the inhibition capacity of the examined compounds to inhibit the enzymes and their correlation to the inhibition results of experimental enzymes, all the compounds from GC-MS profiles of methanolic, *n*-hexane, ethyl acetate, and *n*-butanol fractions were docked against urease and α-glucosidase enzymes, along with hydroxy urea and quercetin (standards) docked against urease and α-glucosidase enzymes. The in silico molecular docking results depict the interaction of urease and α-glucosidase with the ligands benzenepropanoic acid, 3,5-bis(1,1-dimethylethyl)-4-hydroxy-, methyl ester, ar-turmerone, curlone, 3-pyrazolidinone, and 4,4-dimethyl-1-phenyl detected in GC-MS analysis, which conclusively supports our observation of the plant extract in terms of urease and alpha-glucosidase inhibitory assays. The favorable in vitro potential of any extract should always be followed by toxicological experiments to determine the safety level and beneficial effects on animal models and it will be included in future studies [78].

## 4. Materials and Methods

### 4.1. Plant Collection and Identification, and Chemicals

The whole plant of *V. officinalis* was collected during the flowering season from 31°10′35″ N 72°42′13″ E Chak NO. 363 JB, Tehsil Gojra, District Toba Tek Singh, Punjab, Pakistan from November 2017 to March 2018. The taxonomic status of the plant was verified by Botanist Government College University, Lahore, Pakistan. The plant specimen with a voucher number of 3514 was deposited in the Botany department of the university. The solvents and chemicals of methanol, *n*-hexane, ethyl acetate, *n*-butanol, Folin-Ciocalteu reagent, sodium carbonate, gallic acid, aluminum chloride, quercetin, DPPH solution, Trolox, 2,2-azinobis(3-ethylbenothiazoline) 6-sulfonic acid, potassium persulfate, CuCl_2_, Neocuprion, ferric chloride, 2,4,6-tris(2-pyridyl)-s-triazine TPTZ, urease solution, α-glucosidase, and p-nitro-α-D-glucopyranoside were purchased from Sigma-Aldrich Chemical Co Ltd. (Darmstadt, Germany). All other reagents used in the study were of analytical and chromatographic grade. Deionized water was used to prepare all solutions.

### 4.2. Extraction and Fractionation

The whole plant was shade dried and then pulverized into a coarse powder. The pulverized powdered material (10 kg) was then macerated in 80% methanol (20 L) for 2 weeks with frequent shaking at room temperature. Filtration of the methanolic extract was completed using Whatman filter paper and dried in a vacuum under reduced pressure at 40 °C by a rotary evaporator to produce a dry crude methanolic extract. The dry crude methanolic extract (460 g) of *V. officinalis* was suspended in 1000 mL of distilled water. *n*-Hexane (20 g), ethyl acetate (100 g), and *n*-butanol (95 g) were used as extraction solvents to obtain different solvent fractions. Each fraction was then concentrated by using a rotary evaporator, followed by a 45 °C oven dry extraction. All extracts are kept in the refrigerator in air-tight containers for future assessment [77].

### 4.3. Phytochemical Assessment of V. officinalis

#### 4.3.1. Preliminary Phytochemical Assessment

Various phytochemical tests were performed for the phytochemical analysis of *V. officinalis* to evaluate the primary and secondary groups of metabolites in its methanolic extract along with its different fractions. The identification of primary metabolites including carbohydrates, amino acids, proteins, and starch was completed using Molish’s test, Ninhydrin, Biuret test, and Iodine test, respectively. The screening of secondary metabolites such as saponins (Frothing test), tannins (Ferric-Chloride test), phenols (Lead acetate test), flavonoids (Amyl Alcohol test), alkaloids (Dragendroff’s test), glycosides (Erdmann’s test, Borntrager’s Test, Keller-killani test), resin (Acetic-anhydride test) and Steroids and Terpenes (Salkowski’s test) were also done according to standard methods [79].

#### 4.3.2. Estimation of Polyphenolic Contents

Determination of TPC

The TPC of the crude methanolic sample and its different fractions were determined using the Folin-Ciocalteu reagent as reported previously with some modifications [62]. The sample solution of concentration 1 mg·mL^−1^ was made in methanol. The volume of 200 µL of sample solution was mixed with 200 µL of Folin Ciocalteu reagent in a 2 mL test tube and was vigorously mixed by the vortex. Then 0.8 mL of sodium carbonate solution (700 µM) was added to the mixture. The mixture was incubated for 2 h at ambient temperature, followed by the transfer of 200 µL of assay sample mixture to a 96-microtiter plate. The absorbance of each sample was recorded at λ 765 nm by using the instrument Biotek-Synergy HT. The same procedure was completed by producing aliquots of gallic acid’s at various concentrations including 10, 20, 40, 60, 80, 100, and 200 µg·mL^−1^ in methanol and the calibration curve was drawn by recording the absorbance of each aliquot of gallic acid at λ 765 nm. The methanol was used as a negative control. Total phenolic content was expressed in milligrams of gallic acid per gram of dry extract (mg·GAE·g^−1^ DE).

Determination of TFC

The TFC of each sample extract solution, including methanolic and its fractions, was assessed using a modified aluminum chloride method as reported in previous literature [80]. The stock solution for each extract solution had a concentration of 1 mg·mL^−1^, prepared in methanol. A solution mixture was prepared by combining 1 mL of sample extract (1 mg·mL^−1^), 4 mL of deionized water, 300 µL of sodium nitrite solution (5%), and 300 µL of AlCl_3_ solution (10%). Two mL of sodium hydroxide solution (1 M) was added, incubated for 6 min, then 2.4 mL of deionized water was added. The absorbance of each sample mixture solution was measured at λ 510 nm using an instrument UV-visible spectrophotometer IRMECO U2020. The same procedure was repeated by preparing the solution of quercetin’s different aliquots including 50, 100, 200, 300, 400, 500, 600, 800, and 1000 µg·mL^−1^ in methanol. The methanol (solvent) was used as the negative control. The calibration curve of quercetin was acquired by recording the absorbance of each aliquot of quercetin at λ 510 nm. The result of TFC of each sample extract was expressed as milligrams of quercetin per gram of dry extract (mg QE·g^−1^ DE).

#### 4.3.3. GC-MS Analysis

CRVO, NHVO, EAVO, and NBVO fractions of the whole plant of *V. officinalis* were studied by employing GC-MS. GC-MS was conducted using a gas chromatograph (Agilent 7890B) combined with an Agilent 5977B MSD equipped with mass hunter acquisition software. The system consisted of an HP-5ms ultra inert column with dimensions (30 m × 250 µm, 0.25 μm). The carrier gas was helium at a flow rate of 1.3 mL/min in constant flow mode. The temperature at the front inlet was adjusted to 250 °C. The initial oven temperature was held at 50 °C for 2 min, and then the oven temperature steadily increased from 50 °C to 200 °C at a rate of 15 °C/min. The sample extract was prepared in one microliter solution strength and was injected. MS source and MS Quad temperature were set at 230 °C and 151 °C, respectively. The identification was made using a scanning ranging from 50 to 1000 *m*/*z* and metabolites were identified by a comparison of the mass spectrum of each separated metabolite on specific retention time with mass spectrum data stored in the NIST-14 library [81].

### 4.4. Antioxidant Assays

#### 4.4.1. Radical Scavenging Potential

The radical scavenging potential of different extracts of *V. officinalis* was assessed by using DPPH and ABTS assays with minor modifications described in the literature [5].

2,2-diphenyl-1-picrylhydrazyl (DPPH) assay

Each extract was added into a sufficient amount of methanol to acquire the desired concentration of each sample (0.3127 mg·mL^−1^). The 50 µL of each extract solution was added to a 96-microtiter plate followed by the addition of 150 µL 200 mM DPPH solution. The mixture was incubated at room temperature in dark for 30 min. The same procedure was repeated for the different concentrations of Trolox between 5–100 µg·mL^−1^ (positive control) to generate a calibration curve for the calculation of scavenging potential. The same procedure was conducted for the blank (negative control) by adding 50 µL methanol instead of Trolox or sample. The absorbance was measured at λ 517 nm using an instrument Bio Tek Synergy HT reader. The results of antioxidant potential were exhibited in milligram Trolox equivalent per gram of dry extract (mg TE·g^−1^ DE).

2-azino-bis(3-ethylbenzothiazoline-6-sulfonic acid (ABTS) assay

An equal volume of 2,2-azinobis(3-ethylbenothiazoline) 6-sulfonic acid (2.5 mM) and potassium persulfate (2.45 mM) was mixed, and 2 mL of this mixture was added to a sample solution of 1 mL (0.3127 mg·mL^−1^) in a glass test tube. The test tube was incubated for 30 min in the dark and absorbance was noted at 734 nm. To generate a calibration curve for Trolox, the same procedure was repeated using 5–80 µg·mL^−1^ solution of Trolox, with methanol as the negative control. The results were presented in milligram Trolox per gram of dry extract (mg TE·g^−1^ DE).

#### 4.4.2. Reducing Power Antioxidant Assay

The reducing antioxidant potential of different fractions and methanolic extract of *V. officinalis* was evaluated by two methods namely cupric ion reducing antioxidant capacity (CUPRAC) and ferric reducing antioxidant power (FRAP) assays in accordance with the modified procedure reported in the literature [82].

CUPRAC assay

A reaction mixture was prepared by taking the equal volume (1:1:1) of CuCl_2_ 10 mM, neocuprion 7.5 mM, ammonium acetate buffer 1M pH 7, and then 3 mL of this reaction mixture was mixed with a sample solution of the 0.5 mL (0.3127 mg·mL^−1^). The mixture was then incubated at room temperature for 30 min. The solution’s absorbance was noted at λ 450 nm. For blank, methanol was used. The calibration curve was drawn for Trolox by using the concentrations of Trolox between 2.5–100 µg·mL^−1^. The results were presented in mg Trolox per gram of dry extract (mg·TE·g^−1^ of DE).

FRAP assay

A reaction mixture was made by taking acetate buffer (0.3 M, pH 3.6), ferric chloride (20 mM), 2,4,6-tris(2-pyridyl)-s-triazine TPTZ (10 mM) in HCl (40 mM) (10:1:1). Then 2 mL of this reaction mixture was mixed with each extract solution of 30 µL (0.3127 mg·mL^−1^). The resulting mixture was incubated for 30 min at room temperature. The absorbance was determined at 593 nm. For blank, methanol was used. The calibration curve for Trolox was drawn to calculate the antioxidant potential. The results were exhibited in mg Trolox per gram of dry extract (mg·TE·g^−1^ of DE).

### 4.5. In Vitro Enzyme Inhibition Assay

The in vitro biological potential of different extracts of *V. officinalis* was assessed using two significant enzymes i.e., urease and α-glucosidase enzymes.

#### 4.5.1. Urease Enzyme Inhibition Assay

The anti-urease potential of methanolic crude extract and different fractions of *V. officinalis* were evaluated by using the reported method [41]. A mixture of 20 µL of urease solution (0.025%) prepared in phosphate buffer (1 M, pH 7.0) and 20 μL of extract sample was added in a microtiter plate and then kept for incubation for 15 min at room temperature. Then 60 µL of aqueous urea solution (2.25%) was mixed with the resultant reaction mixture and kept for incubation for 15 min at room temperature and absorbance was recorded at 630 nm (pre-read). Then 60 µL of phenol reagent and 100 µL of solution of sodium hypochlorite (prepared in alkali) were added to the above reaction mixture which was then incubated at room temperature for 30 min. The absorbance was noted at 630 nm (after read). For positive control, hydroxy urea and for negative control phosphate buffer was used. The % inhibition of the urease enzyme was determined using the following Equation (1).
Inhibition activity (%) = 1 − (A_sample_/A_control_) × 100(1)

A_sample_—absorbance of sample; A_control_—absorbance of control.

#### 4.5.2. α-Glucosidase Enzyme Inhibition Assay

The α-glucosidase enzyme inhibition activity has been conducted in accordance with a previously reported modified method [81]. A mixture of a solution of enzyme 10 µL (1 U/mL), 50 µL of phosphate buffer (50 mM, pH 6.8) and 20 µL of the sample solution was added in 96 well microtiter plate and incubated for 15 min at room temperature. The absorbance was taken at 405 nm (pre-read). The volume of 20 µL of substrate solution of p-nitro-α-D-glucopyranoside (0.5 mM) was added to the above reaction mixture solution and then kept in incubation again for 15 min at room temperature. The absorbance was measured at 405 nm (after read). The method was repeated with quercetin (positive control) and methanol (negative control). The % inhibition of the enzyme was computed using Equation (1).

### 4.6. Hemolytic Activity

Using the method previously described, the hemolytic activity of extract/fractions obtained from plants was assessed [83]. In a sterile screw top EDTA tube, 10 mL of human blood from volunteers was added. The tube was then centrifuged at 850 g for 5 min. The top portion was removed, and erythrocytes were then repeatedly washed using 10 mL of cold, sterile, isotonic PBS (Phosphate-Buffered Saline) at a pH of 7.4. In 20 mL of sterile, cold PBS, the washed cells were once again suspended. Erythrocyte solution was mixed with the extracts (1000 µg/1 mL) and incubated at 37 °C for 1 h. The hemolysis rate was calculated using the hemoglobin absorbance in the supernatant at 540 nm. PBS was employed as the negative control, and the positive control was 0.1 percent Triton X-100. The following Equation (2) was used to assess the hemolysis percentage.
Hemolysis (%) = (A_sample_ − A_negative control_)/A_positive control_ × 100(2)

A_sample_—absorbance of sample; A_negative control_—absorbance of negative control; A_positive control_—absorbance of positive control

### 4.7. Molecular Docking

Several tools, including Auto Dock vina software, MGL Tools, Discovery Studio, PyRx, and Babel, were utilized for molecular docking. Using the Discovery Studio, the receptor molecule that was downloaded from the protein data library [84] was further prepared for increasing the efficacy of enzymes [85]. The Babel was used to prepare ligand compounds. These produced ligands and receptors were uploaded into Vina, which was built into PyRx. Finally, Vina was used for docking. Discovery Studio was used to visualize the results [77].

### 4.8. Statistical Analysis

The tests were presented in triplicates. The findings were exhibited as a mean of triplicate ± standard deviation. One-way ANOVA was performed with IBM SPSS statistics 23 by applying Post Hoc Tukey’s Test. *p* ≤ 0.05 values remained set as a significant value.

## 5. Conclusions

The current study analyzed *in-vitro* antioxidant potential, urease and α-glucosidase enzyme inhibition activity and hemolytic potential of the whole plant of *V. officinalis* fractions. Ethyl acetate fraction in comparison to other fractions showed the maximum polyphenolic contents (TFC and TPC) which correlate with the current results of antioxidant, urease and α-glucosidase activities of this plant. Moreover, bioactive compounds identified by GC-MS in all fractions of *V. officinalis* also validated the results of this study. The urease and α-glucosidase inhibition activities of *V. officinalis* were further justified by in silico molecular docking studies of GC-MS-identified ligands, Benzenepropanoic acid and 3,5-bis(1,1-dimethylethyl)-4-hydroxy-, methyl ester, ar-Turmerone, Curlone, 3-pyrazolidinone, 4,4-dimethyl-1-phenyl with these enzymes. The biological and phytochemical potential of this plant demonstrated its importance for the ongoing process of further isolating bioactive compounds.

## Figures and Tables

**Figure 1 molecules-27-06685-f001:**
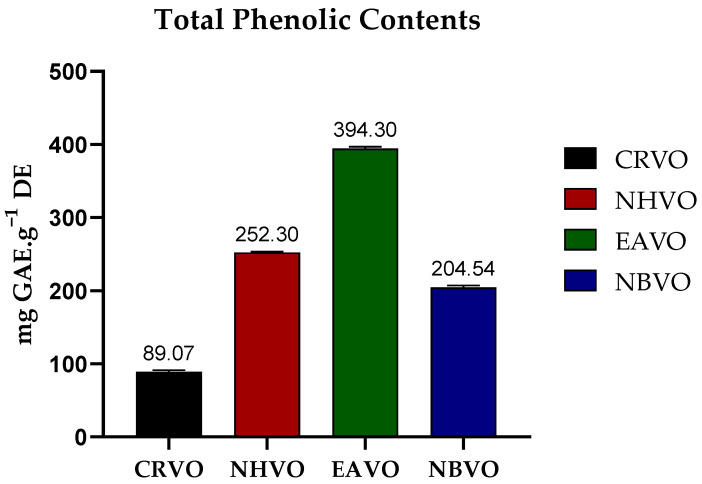
Total phenolic content (TPC) of the whole plant of *Verbena officinalis* fractions (All experiments were performed in triplicates and the error bar represents standard deviation). CRVO; crude methanol fraction, NHVO; *n*-hexane fraction, EAVO; ethyl acetate fraction, NBVO; *n*-butanol fraction, GAE; gallic acid equivalent and DE; dry extract.

**Figure 2 molecules-27-06685-f002:**
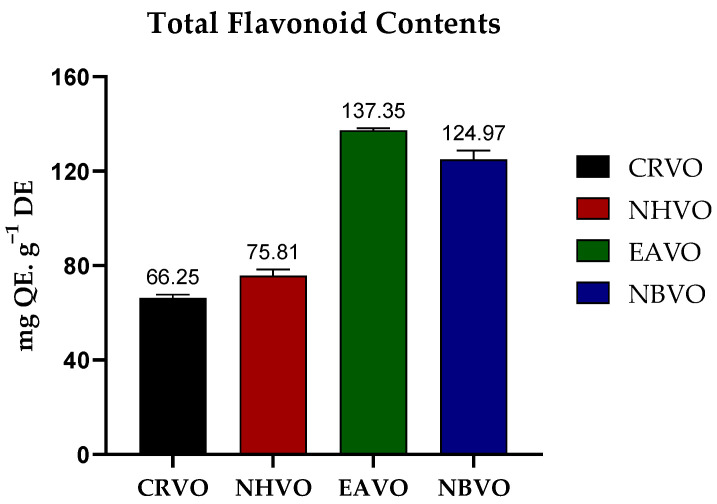
Total flavonoid content (TFC) of the whole plant of *Verbena officinalis* fractions (All experiments were performed in triplicates and the error bar represents standard deviation). CRVO; crude methanol fraction, NHVO; *n*-hexane fraction, EAVO; ethyl acetate fraction, NBVO; *n*-butanol fraction, QE; quercetin equivalent and DE; dry extract.

**Figure 3 molecules-27-06685-f003:**
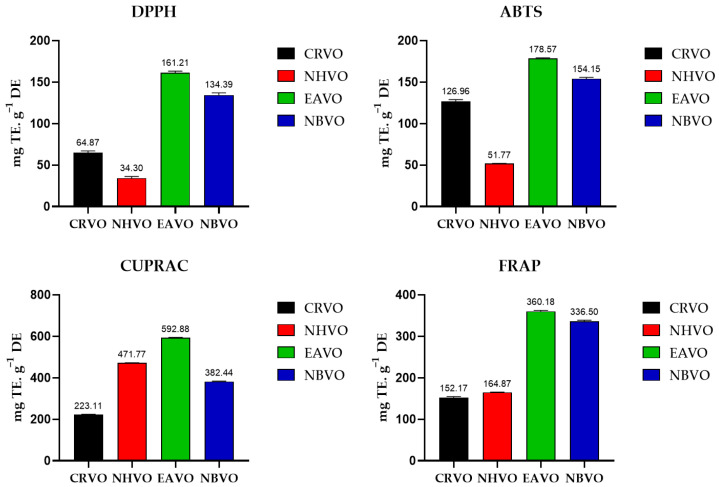
Antioxidant activity of crude methanolic extract (CRVO) and different fractions (*n*-hexane extract (NHVO), ethyl acetate extract (EAVO), and *n*-butanol extract (NBVO)) of the whole plant of *Verbena officinalis* by DPPH ABTS, CUPRAC, and FRAP assays. (All experiments were performed in triplicates, and the error bar represents the standard deviation).

**Figure 4 molecules-27-06685-f004:**
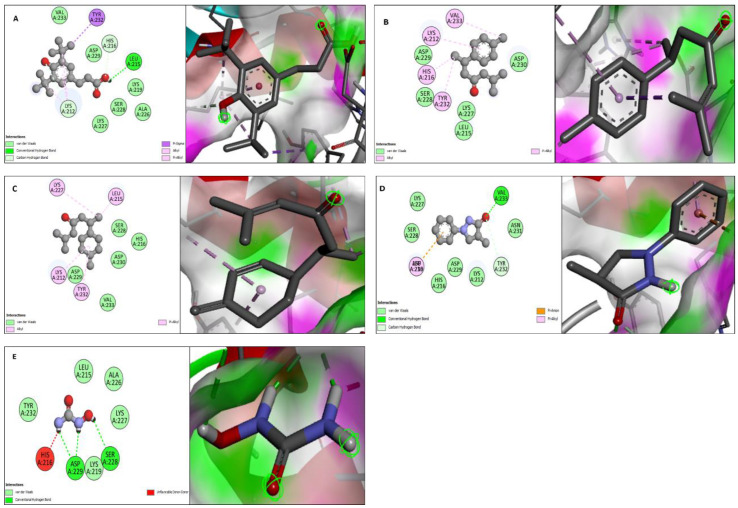
2D and 3D interaction of urease with ligands. (**A**) Benzenepropanoic acid, 3,5-bis(1,1-dimethylethyl)-4-hydroxy-, methyl ester, (**B**) ar-Turmerone, (**C**) Curlone, (**D**) 3-pyrazolidinone, 4,4-dimethyl-1-phenyl and (**E**) Hydroxy urea.

**Figure 5 molecules-27-06685-f005:**
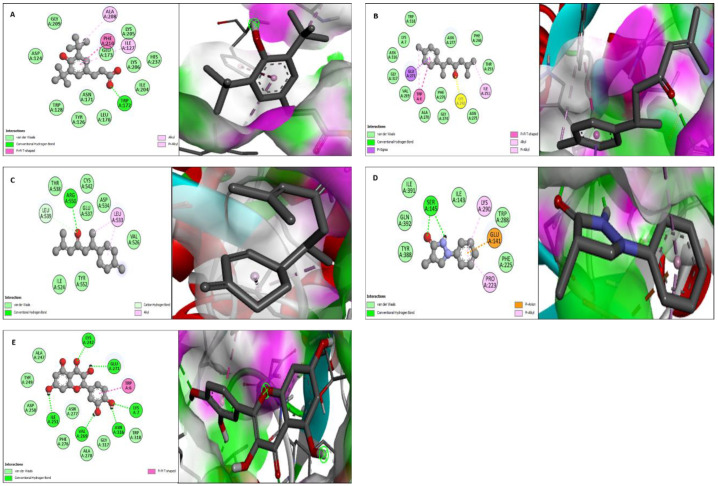
2D and 3D interaction of α-glucosidase with ligands. (**A**) Benzenepropanoic acid, 3,5-bis(1,1-dimethylethyl)-4-hydroxy-, methyl ester, (**B**) ar-Turmerone, (**C**) Curlone, (**D**) 3-pyrazolidinone, 4,4-dimethyl-1-phenyl and (**E**) Quercetin.

**Table 1 molecules-27-06685-t001:** Preliminary phytochemical assessment of the methanolic crude extract of *Verbena officinalis* and its different fractions.

No.	Class of Metabolites	Test Name	CRVO	NHVO	EAVO	NBVO
1	Carbohydrate	Molish’s test	+	−	+	+
2	Amino acid	Ninhydrin test	−	−	−	−
3	Protein	Biuret test	−	−	−	−
4	Saponin	Frothing test	+	+	+	+
5	Tannin	Ferric-chloride test	+	−	+	+
6	Phenol	Lead acetate test	+	+	+	+
7	Flavonoids	Amyl alcohol test	+	+	+	+
8	Starch	Iodine test	+	+	+	+
9	Alkaloid	Dragendroff’s test	+	+	+	+
10	Glycosides	Erdmann’s test	+	−	−	−
		Borntrager’s test	−	−	−	−
		Keller-killani test	+	−	+	+
11	Resins	Acetic-anhydride test	+	+	+	+

CRVO; crude methanol fraction, NHVO; *n*-hexane fraction, EAVO; ethyl acetate fraction, NBVO; *n*-butanol fraction, +; present and −; absent.

**Table 2 molecules-27-06685-t002:** Urease and α-glucosidase inhibition values of the methanolic crude extract of *Verbena officinalis* and its different fractions.

Sample Fraction	Urease IC_50_ (µg·mL^−1^)	α-Glucosidase IC_50_ (µg·mL^−1^)
CRVO	465 ± 20.20 ^A^	NA
NHVO	324 ± 16.40 ^B^	420 ± 20 ^B^
EAVO	10 ± 1.60 ^D^	685 ± 31 ^A^
NBVO	30 ± 2.40 ^C^	NA
Standard	9.8 ± 1.20 * ^D^	10 ± 1.30 ** ^C^

All tests were conducted in triplicates and results were expressed as mean ± S.D (The results of all samples significantly vary *p* ≤ 0.05). ^A,B,C,D^ Values with the different superscript letters (within a column) are significantly different. *; Hydroxyurea, **; Quercetin, CRVO; crude methanol extract, NHVO; *n*-hexane extract, EAVO; ethyl acetate extract, NBVO; *n*-butanol extract, and NA; no activity.

**Table 3 molecules-27-06685-t003:** Hemolytic potential of crude methanolic extract (CRVO) and different fractions (*n*-hexane extract (NHVO), ethyl acetate extract (EAVO) and *n*-butanol extract (NBVO)) of *Verbena officinalis*.

Sample Fraction	Hemolytic Activity (%)
CRVO	6.5 ± 0.94 ^E^
NHVO	7.2 ± 0.85 ^D^
EAVO	10.1 ± 1.30 ^C^
NBVO	14.5 ± 1.20 ^B^
Triton X-100	93.5 ± 0.48 ^A^

All tests were conducted in triplicates and results were expressed as mean ± S.D. (The results of all samples significantly vary by *p* ≤ 0.05). ^A,B,C,D,E^ Values with the different superscript letters (within a column) are significantly different. CRVO; crude methanol extract, NHVO; *n*-hexane extract, EAVO; ethyl acetate extract, and NBVO; *n*-butanol extract.

**Table 4 molecules-27-06685-t004:** Molecular docking of urease and α-glucosidase with different ligands representing binding affinity and interacting amino acids.

No.	Name of Compounds	Urease (Binding Affinity Kcal.·mol^−1^)	Interacting Amino Acid Residues	α-Glucosidase (Binding Affinity Kcal.·mol^−1^)	Interacting Amino Acid Residues
1	Benzenepropanoic acid, 3,5-bis(1,1-dimethylethyl)-4-hydroxy-, methyl ester	−6.8	Lys212, Leu215, His216, Lys219, Ala226, Lys227, Ser228, Asp229, Tyr232, Val233	−6.8	Asp124, Tyr126, Ile127, Trp128, Leu170, Asn171, Trp172, Glu173, Ile204, Lys205, Lys206, Ala208, Gly209, Phe210, His237
2	ar-Turmerone	−5.8	Lys212, Leu215, His2016, Lys227, Ser228, Asp229, Asp230, Tyr232, Val233	−6.5	Trp6, Lys7, Lys242, Ile251, Thr253, Val269, Ala270, Glu271, Gly274, Asn275, Phe276, Asn277, Asn316, Gly317, Trp318
3	Curlone	−5.6	Lys212, Leu215, His216, Lys227, Ser228, Asp229, Asp230, Tyr232, Val233	−5.9	Ile524, Val526, Leu533, Asp534, Glu537, Thr538, Leu539, Cys542, Arg550, Tyr552
4	3-pyrazolidinone, 4,4-dimethyl-1-phenyl	−5.7	Lys212, His216, Lys227, Ser228, Asp229, Asp230, Asn231, Tyr232, Val233	−5.8	Glu141, Ile143, Ser145, Pro223, Phe225, Trp288, Lys90, Tyr388, Ile391, Gln392
5	(Standard)	−4.1 *	Leu215, His216, Lys219, Ala226, Lys227, Ser228, Asp229, Tyr232	−7.9 **	Trp6, Lys7, Lys242, Ala247, Tyr249, Asp250, Ile251, Val269, Ala270, Glu271, Phe276, Asn277, Asn316, Gly317, Trp318

* Hydroxy urea and ** Quercetin.

## Data Availability

All data are available in the text.

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
