# Peer review of "Metabolic Profiling by GC-MS, In Vitro Biological Potential, and In Silico Molecular Docking Studies of Verbena officinalis"

_molecules, 2022, doi:10.3390/molecules27196685_

Round 1

Reviewer 1 Report

English should be revised throughout

Acronyms/Abbreviations should be defined the first time (and only one time) they appear in each of three sections: the abstract; the main text; the first figure or table (when defined for the first time, the acronym/abbreviation should be added in parentheses); then use acronyms 

- “Verbena officinalis L.” when first used in Abstract (Line 24) and text (Line 87)

- “mg GAE.g-1 DE” instead of “mg GA.Eq.gm-1 DE” throughout including figures/tables

- “mg QE.gm-1 DE” instead of “mg Q.Eq.gm-1 DE throughout including figures/tables

- “mg TE.g-1 DE” instead of “mg T.Eq.gm-1 DE” throughout including figures/tables

- “g” instead of “gm” throughout

Lines 32-33: “... by all four antioxidant methods.” instead of “... by four antioxidant methods (DPPH, ABTS, CUPRAC, and FRAP).” (they were mentioned before)

Lines 39-40: “... 24, 56, 25, and 9 bioactive compounds, respectively, with 80% quality index.”

Line 44: “many” can be deleted

Line 45: “GC-MS”

Lines 58-59: “... of therapeutic uses, including antioxidant, antibacterial, anti-inflammatory, antiviral, antifungal, and anticancer” 

Line 68: “... disorders, such as duodenal...

Line 72: “nowadays 

Lines 81-82: “Several inhibitors of α-glucosidase, such as miglitol and acarbose, have been discovered [16]. However, acarbose use ...”

Line 88: “all over Europe”

Statistical data should be presented in Results/Figures

Line 132-133: text goes after Line 134; “and Qu; quercetin” should be replaced by “GA; gallic acid

Lines 142-143: text goes after Line 144; “and “GA; gallic acid” should be replaced by “Q; quercetin”

Lines 152-153: “retention time (RT) expressed in minutes

Line 192 (Figure 7) and Lines 196-197 (Table 6): “CUPRAC” not “CUPARAC”

Line 226: a little overstated to use “ with promising antidiabetic potential” when compared to quercetin 

Line 231: “percentage” instead of “%age

Lines 282-283: please revise sentence 

Line 298: “were reported”

Lines 305-306: please add numbers when discussing previous studies

Lines 308-313: sentences need references

Lines 316-318: to improve the quality of the manuscript, some more information (how natural antioxidants can prevent the oxidation process?) can be added here, such as “bioactive compounds from plants exert their antioxidant activity via multiple mechanisms, including activation of Nrf2/ARE and deactivation of the NF-kB pathways, directly involved in the inflammatory reaction” (doi: 10.3390/antiox11071412)

Line 372: “excellent potent inhibitor “?

Line 391: the sentence “Plant hemolytic activity was expressed in % hemolysis.” can be deleted; it belongs to Methods

Lines 308-337: please discuss how the results from the mentioned studies compare to your results

Lines 338-364: at the end of the paragraph you state that “there is no comprehensive research reported on the urease inhibitory activity of whole plant of V. officinalis.” Please compare the results of this plant with some other plants

Lines 365-380: same here 

At the end of Discussion the following idea should be added: “the favorable in vitro potential of any extract should always be followed by toxicological experiments to determine the safety level and beneficial effects on animal models” (Vedeanu et al. Doi: 10.1071/EN19249) 

Line 423: please mention geographical coordinates

In Materials and Methods: 

- the source of all reagents and all devices used should be mentioned

- references should be used for all protocols

- all equations should be written in mathematical form first

References should be checked (see # 23)

Author Response

Response to Reviewer 1 Comments

Dear reviewer,

We greatly appreciated your careful reading of our manuscript and your valuable comments, which enormously allowed us to improve our manuscript quality. We have carefully considered the comments and have revised and rewritten the manuscript accordingly. Thank you very much for your time and the careful review. The revised manuscript was marked up with the changed text by highlighting them in yellow.

Comments and Suggestions for Authors

English should be revised throughout

Response: The authors are thankful to the reviewer for generating very valuable comments for the improvement of the manuscript. We have read all the manuscript thoroughly and English have also been improved and revised by native English speaker.

Acronyms/Abbreviations should be defined the first time (and only one time) they appear in each of three sections: the abstract; the main text; the first figure or table (when defined for the first time, the acronym/abbreviation should be added in parentheses); then use acronyms.

Response: The authors are extremely thankful to the reviewers and editor for sparing their quality time for the improvement of our manuscript. As you suggested, the abbreviations have been revised in the whole manuscript.

- “Verbena officinalis L.” when first used in Abstract (Line 24) and text (Line 87)

Response: The required change in the manuscript has been performed.

- “mg GAE. g-1 DE” instead of “mg GA. Eq. gm-1 DE” throughout including figures/tables

- “mg QE. g-1 DE” instead of “mg Q. Eq. gm-1 DE” throughout including figures/tables

- “mg TE. g-1 DE” instead of “mg T. Eq .gm-1 DE” throughout including figures/tables

- “g” instead of “gm” throughout

Response: Thanks for your comment. These units have been modified throughout the manuscript.

Lines 32-33: “...by all four antioxidant methods.” instead of “...by four antioxidant methods (DPPH, ABTS, CUPRAC, and FRAP).” (they were mentioned before)

Response: The authors acknowledge the reviewer’s expertise and changes have been performed accordingly.

Lines 39-40: “…24, 56, 25, and 9 bioactive compounds, respectively, with 80% quality index.”

Response: Thanks for your suggestion. As suggested, this sentence has been improved.

Line 44: “many” can be deleted

Response: Thanks for your comment. “Many” has been deleted.

Line 45: “GC-MS”

Response: Thanks for your comment. The change has been performed in the manuscript.

Lines 58-59: “...of therapeutic uses, including antioxidant, antibacterial, anti-inflammatory, antiviral, antifungal, and anticancer”

Response: The advised change is performed in the manuscript.

Line 68: “...disorders, such as duodenal...”

Response: The advised change is performed in the manuscript.

Line 72: “nowadays”

Response: The advised change is performed in the manuscript.

Lines 81-82: “Several inhibitors of α-glucosidase, such as miglitol and acarbose, have been discovered [16].

Response: The advised change is performed in the manuscript.

However, acarbose use...”

Response: The advised change is performed in the manuscript.

Line 88: “all over Europe”

Response: The advised change is performed in the manuscript.

Statistical data should be presented in Results/Figures

Response: Thanks for your suggestion. As suggested, statistical data are added to the figures.

Line 132-133: text goes after Line 134; “and Qu; quercetin” should be replaced by “GA; gallic acid”

Response: The authors are thankful for highlighting our mistake and for improving the mistake. This has been corrected.

Lines 142-143: text goes after Line 144; “and “GA; gallic acid” should be replaced by “Q; quercetin”

Response: The authors are thankful for highlighting our mistake and for improving the mistake. This has been corrected.

Lines 152-153: “retention time (RT) expressed in minutes”

Response: Thanks for your suggestion. As suggested, this sentence is modified.

Line 192 (Figure 7) and Lines 196-197 (Table 6): “CUPRAC” not “CUPARAC”

Response: Thanks for your comment. These corrections have been done in the manuscript.

Line 226: a little overstated to use “with promising antidiabetic potential” when compared to quercetin

Response: Thanks for your comment. This sentence has been corrected.

Line 231: “percentage” instead of “%age”

Response: Thanks for your comment. This has been modified.

Lines 282-283: please revise sentence

Response: Thanks for your suggestion. As suggested, the sentence has been rephrased and highlighted.

Line 298: “were reported”

Response: Thanks for your suggestion. As suggested, the line has been modified accordingly.

Lines 305-306: please add numbers when discussing previous studies

Response: Thanks for your suggestion. As suggested, the values of flavonoids and phenolics are added for hydroalcoholic and aqueous extracts.

Lines 308-313: sentences need references

Response: Thanks for your comment. The citation is inserted in the described sentence.

Lines 316-318: to improve the quality of the manuscript, some more information (how natural antioxidants can prevent the oxidation process?) can be added here, such as “bioactive compounds from plants exert their antioxidant activity via multiple mechanisms, including activation of Nrf2/ARE and deactivation of the NF-kB pathways, directly involved in the inflammatory reaction” (doi: 10.3390/antiox11071412)

Response: The authors acknowledge the expertise of a respected reviewer. The mechanism along with citation is inserted in the manuscript.

Line 372: “excellent potent inhibitor“?

Response: Thanks for your comment. The sentence has been revised.

Line 391: the sentence “Plant hemolytic activity was expressed in % hemolysis.” can be deleted; it belongs to Methods

Response: Thanks for your suggestion. As suggested, these sentences have been omitted.

Lines 308-337: please discuss how the results from the mentioned studies compare to your results

Response: Thanks for your suggestion. As suggested, the results of this part have been revised.

Lines 338-364: at the end of the paragraph, you state that “there is no comprehensive research reported on the urease inhibitory activity of whole plant of V. officinalis.” Please compare the results of this plant some other plants

Lines 365-380: same here

Response: Thanks for your suggestion. As suggested, the results of other plants for inhibition of urease and glucosidase have been added and highlighted.

At the end of Discussion the following idea should be added: “the favorable in vitro potential of any extract should always be followed by toxicological experiments to determine the safety level and beneficial effects on animal models” (Vedeanu et al. Doi: 10.1071/EN19249)

Response: The authors acknowledged this idea and incorporated it into the text for future studies.

Line 423: please mention geographical coordinates

Response: Thanks for your suggestion. As suggested, the geographical coordinates have been added.

In Materials and Methods:

- the source of all reagents and all devices used should be mentioned

Response: Response: Thanks for your suggestion. As suggested, the chemical and reagents have been revised and mentioned.

- references should be used for all protocols

Response: The references for all protocols have been added.

- all equations should be written in mathematical form first

Response: The suggestion is followed accordingly.

References should be checked (see # 23)

Response: All the references have been checked again and reference 23 (in revised manuscript 25) is corrected.

Finally, we would like to express our sincere thanks for the appreciated comments and suggestions that helped us enhance our manuscript’s quality.

Reviewer 2 Report

This paper is potentially interesting but there are some issues that should be carefully addressed by authors before making the paper suitable for publication in the Molecules.

In abstract, please describe the samples (crude methanolic extract fractioned….)

Line 94: diuretics and expectorant belong more to bioactivities than in the ailments

Figures 3-6: These chromatograms are not very representative (e.g. significant baseline drift and poor resolving power for some compounds.

Line 153: Please define %Area

Table 6 and Figure 7: Please use table or figure, not both for presenting the same results.

Line 222: If you tested differences statistically, please indicate significant differences in Table 7.

Please explain sign ‘’-‘’ in Table 7.

Discussion: The text should not repeat table data but emphasize or summarize the most important observations. In the section relating to TPC and TFC it might be useful to compare/discuss the results with other authors, and highlight the phenolics that are the most important for antioxidant activity.

Line 428: Please add mass of the sample and volume of used solvents. Also, please add extraction yield.

Line 470: Two mL, not 2 mL at the beginning of the sentence.

Line 491: Please add scan range (m/z).

Line 511: What was the blank?

Lines 571-573: Please delete duplicate procedure.

Line 594: Which post hoc test?

Studies involving humans should include a statement on ethics approval.

Author Response

Response to Reviewer 2 Comments

Dear reviewer,

We greatly appreciated your careful reading of our manuscript and your valuable comments, which enormously allowed us to improve our manuscript quality. We have carefully considered the comments and have revised and rewritten the manuscript accordingly. Thank you very much for your time and the careful review. The revised manuscript was marked up with the changed text by highlighting them in green.

Comments and Suggestions for Authors

This paper is potentially interesting but there are some issues that should be carefully addressed by authors before making the paper suitable for publication in the Molecules.

Response: Thank you very much for your positive feedback. We hope the following corrections and responses to your comments and suggestions satisfy you requests.

In abstract, please describe the samples (crude methanolic extract fractioned….)

Response: Thanks for your comment. The information has been revised in the abstract.

Line 94: diuretics and expectorant belong more to bioactivities than in the ailments

Response: The authors acknowledge the expertise of the reviewer, and this has been corrected now.

Figures 3-6: These chromatograms are not very representative (e.g., significant baseline drift and poor resolving power for some compounds.

Response: The authors are thankful to the reviewer for giving suggestions for the improvement of this manuscript. The problem is that we don’t have a GC-MS facility in campus. We have performed GC-MS on commercial. They provided us spectra as pdf files. We took a screenshot from those files. That is why we cannot improve the resolution of these spectra. In addition, we tried our best to improve the quality of the chromatograms.

Line 153: Please define % Area

Response: Thanks for your suggestion. As suggested, the information has been added.

Table 6 and Figure 7: Please use table or figure, not both for presenting the same results.

Response: Thanks for your suggestion. As suggested, we have omitted the repetition in the form of a table. And the results are in the graphs.

Line 222: If you tested differences statistically, please indicate significant differences in Table 7.

Response: Thanks for your suggestion. As suggested, the significant differences have been included.

Please explain sign ‘’-‘’ in Table 7.

Response: Thanks for your suggestion. As suggested, the explanation has been added.

Discussion: The text should not repeat table data but emphasize or summarize the most important observations. In the section relating to TPC and TFC it might be useful to compare/discuss the results with other authors, and highlight the phenolics that are the most important for antioxidant activity.

Response: The authors are thankful to the reviewer for giving such a nice suggestion for the improvement of this manuscript and future studies also. We have discussed the results and made a justification for these results.

Line 428: Please add mass of the sample and volume of used solvents. Also, please add extraction yield.

Response: Thanks for your comment. All the detail required have been mentioned in the manuscript and highlighted.

Line 470: Two mL, not 2 mL at the beginning of the sentence.

Response: Thanks for your comment. This line has been corrected.

Line 491: Please add scan range (m/z).

Response: Thanks for your comment. A scanning range has been added.

Line 511: What was the blank?

Response: Thanks for your comment. The blank sample for each activity has been mentioned.

Lines 571-573: Please delete duplicate procedure.

Response: Thanks for your comment. The duplicate procedure has been deleted.

Line 594: Which post hoc test?

Response: Thanks for your comment. The type of post hoc test has been described in the text.

Studies involving humans should include a statement on ethics approval.

Response: The ethical committee approval statement has been added to the text.

Finally, we would like to express our sincere thanks for the appreciated comments and suggestions that helped us enhance our manuscript’s quality.

Round 2

Reviewer 1 Report

The authors addressed the questions and suggestions. However, there are some details to be attended to:

As the whole plant was used in biological assays, Table 2 can go to Supplemental Materials

Line 214 (Table 3) and Lines 232-234 (Table 4): There are superscript letters (A, B, C, ...); please explain the meaning of these letters

Line 568 (Eq. 1): please correct and add parentheses as following:

Inhibition activity (%) = 1- (Asample/Acontrol) × 100 

Asample - absorbance of sample; Acontrol - absorbance of control

Line 580: it is the same equation as above, so there is no Eq. 2. Line 580 can be deleted). On Lines 578-579, you can say that “The % inhibition of the enzyme was computed using Eq. 1.

Line 591: “Eq. 3” becomes “Eq. 2”

Lines 592-593: please use and describe equation as Eq. 1. See above (Line 568)

Author Response

Response to Reviewer 1 Comments

Dear reviewer,

We greatly appreciated your careful reading of our manuscript and your valuable comments, which enormously allowed us to improve our manuscript quality. We have carefully considered the comments and have revised and rewritten the manuscript accordingly. Thank you very much for your time and the careful review. The revised manuscript was marked up with the changed text by highlighting them in yellow.

Comments and Suggestions for Authors

The authors addressed the questions and suggestions. However, there are some details to be attended to:

As the whole plant was used in biological assays, Table 2 can go to Supplemental Materials

Response: The authors are thankful to the reviewer for generating very valuable comments for the improvement of the manuscript. As suggested Table 2 has been moved to the supplemental materials file.

Line 214 (Table 3) and Lines 232-234 (Table 4): There are superscript letters (A, B, C, ...); please explain the meaning of these letters

Response: Thanks for your comment. As suggested, the superscript letters have been explained.

Line 568 (Eq. 1): please correct and add parentheses as following:

Inhibition activity (%) = 1- (A sample / A control) × 100

A sample - absorbance of sample; A control - absorbance of control

Response: Thanks for your suggestion. As suggested, the correction has been done.

Line 580: it is the same equation as above, so there is no Eq. 2. Line 580 can be deleted). On Lines 578-579, you can say that “The % inhibition of the enzyme was computed using Eq. 1.”

Response: Thanks for your suggestion. As suggested, the correction has been done.

Line 591: “Eq. 3” becomes “Eq. 2”

Response: Thanks for your suggestion. As suggested, the correction has been done.

Lines 592-593: please use and describe equation as Eq. 1. See above (Line 568)

Response: Thanks for your suggestion. As suggested, the correction has been done.

Finally, we would like to express our sincere thanks for the appreciated comments and suggestions that helped us enhance our manuscript’s quality.

Reviewer 2 Report

The manuscript is improved. I suggest deleting Figure 3 due to low quality of chromatograms.

Author Response

Response to Reviewer 2 Comments

Dear reviewer,

We greatly appreciated your careful reading of our manuscript and your valuable comments, which enormously allowed us to improve our manuscript quality. We have carefully considered the comments and have revised and rewritten the manuscript accordingly. Thank you very much for your time and the careful review.

Comments and Suggestions for Authors

The manuscript is improved. I suggest deleting Figure 3 due to low quality of chromatograms.

Response: Thank you very much for your positive feedback. As suggested, Figure 3 has been deleted from the manuscript and put in the supplemental materials file.

Finally, we would like to express our sincere thanks for the appreciated comments and suggestions that helped us enhance our manuscript’s quality.
